# Pencil graphite as electrode platform for free chlorine sensors and energy storage devices

**Jahidul Islam[1], Han Shao[2], Md. Mizanur Rahman Badal[1], Kafil M. Razeeb[2]\*, Mamun Jamal[1]\***

**1** Department of Chemistry, Faculty of Civil Engineering, Khulna University of Engineering & Technology, Khulna, Bangladesh, **2** Micro-Nano Systems Centre, Tyndall National Institute, University College Cork, Cork, Ireland

\* mamun.jamal@chem.kuet.ac.bd (MJ); kafil.mahmood@tyndall.ie (KMR)

**Data Availability Statement:** All relevant data are within the manuscript and its Supporting information files.

**Funding:** MJ: Obtained fund from Ministry of Science & Technology and University Grant

## Abstract

Multifunctional and low-cost electrode materials are desirable for the next-generation sensors and energy storage applications. This paper reports the use of pencil graphite as an electrode for dual applications that include the detection of free residual chlorine using electro-oxidation process and as an electrochemical energy storage cathode. The pencil graphite is transferred to cellulose paper by drawing ten times and applied for the detection of free residual chlorine, which shows a sensitivity of 27 µA mM$^{-1}$ cm$^{-2}$ with a limit of detection of 88.9 µM and linearity up to 7 mM. The sample matrix effect study for the commonly interfering ions such as $NO_3^-$, $SO_4^{2-}$, $CO_3^{2-}$, $Cl^-$, $HCO_3^-$ shows minimal impact on free residual chlorine detection. Pencil graphite then used after cyclic voltammogram treatment as a cathode in the aqueous Zn/Al-ion battery, showing an average discharge potential plateau of ~1.1 V, with a specific cathode capacity of ~54.1 mAh g$^{-1}$ at a current of 55 mA g$^{-1}$. It maintains ~95.8% of its initial efficiency after 100 cycles. Results obtained from the density functional theory calculation is consistent with the electro-oxidation process involved in the detection of free residual chlorine, as well as intercalation and de-intercalation behavior of $Al^{3+}$ into the graphite layers of Zn/Al-ion battery. Therefore, pencil graphite due to its excellent electro-oxidation and conducting properties, can be successfully implemented as low cost, disposable and green material for both sensor and energy-storage applications.

## Introduction

In modern society, the proliferation of portable electronics require highly efficient electrochemical devices such as sensors [1], batteries [2] and supercapacitors [3]. Thereby, designing cost-effective electrodes or electrocatalyst materials with multiple functionalities possesses great potential for the realization of next generation environmental monitoring system [4]. However, considering the cost, availability, and ease of manufacturability, pencil graphite has tremendous potential as an efficient electrode platform for multipurpose applications. Materials based on carbon, such as glassy carbon [5], carbon paste [6], and screen-printed electrodes [7], along with their micro-nano-architectures, are widely used for various electrochemical

Commission of Bangladesh under special allocation programme (2019-2020). KMR: Obtained fund from European Union's Horizon 2020-funded project under grant agreement no. 825114 (SmartVista). The funders had no role in study design, data collection and analysis, decision to publish, or preparation of the manuscript.

**Competing interests:** No authors have competing interests. The authors have declared that no competing interests exist.

applications. Among them, graphite, owing to its outstanding electrochemical properties, has received attention for the improvement of sensors [8] and batteries [9]. However, many of the graphitic materials, such as natural graphite and pyrolytic graphite, exhibit lower rate capabilities at given current densities. Many alternative sources of carbon-based electrodes e.g., pencil graphite leads have been investigated in recent years [10, 11]. The pencil lead is a composite material that is a mixture of graphite, wax, and clay, the proportions of which give the pencil electrode various properties [12]. The possibility of using pencils to draw electrode probes and various other electronic components on an acceptable substrate has only recently been documented [13]. In the majority of cases, pencil graphite electrode (PGE) is modified with various functional groups that possess hindrance in transferring PGE to paper. Moreover, researchers are emphasizing on using electrode materials to participate directly with the analyte, which would make the manufacturing process simple, cost-effective and environmental-friendly. Thereby, in this work PGE has been implemented to detect free residual chlorine (FRC) in aqueous media without any modification.

Disinfection of pathogenic microorganisms in portable water is of vital importance to the safety of public health. Chlorination is a cheap and effective water treatment process; and waterborne disease has been mainly regulated by chlorination in many parts of the world. It is an integral part of traditional drinking water treatment methods to avoid the spread of bacteria in water delivery systems. Sodium hypochlorite, which is the key ingredient for chlorination, undergoes hydrolysis in water to form free residual chlorine consisting of hypochlorous acid (HOCl) and hypochlorite ion (ClO⁻) [14]. Hypochlorite (ClO⁻) is an extremely strong oxidizing agent which acts as a disinfectant. ClO⁻ also used in the bleaching and processing industries. At pH > 8.5 free chlorine exists exclusively as ClO⁻ and at pH < 5.5 free chlorine exists as HOCl (*Eqs* 1 *and* 2) [15]. Growing public understanding of water quality and tougher public health legislation and procedures, such as point-of-use sampling and analysis resulted an increased requirement of developing a robust, accurate, low-cost and portable free chlorine sensor. This is especially true in small and remote communities where highly qualified persons may not be available and routine maintenance is less feasible. Various analytical techniques are existing for the quantification of FRC, including colorimetry, iodometry and UV spectroscopy [15, 16]. However, the electroanalytical method [17] is appealing in this case due to the electroactivity of free chlorine, as well as its simplicity to use. Several promising materials have recently been investigated for FRC sensing with a linear response [17–22]. However, sensing materials they used, are either costly (e.g. glassy carbon, gold, boron doped diamonds, graphene, carbon nanotubes, ferrocene) or potentially dangerous materials (e.g. benzethonium chloride, aniline oligomers). To avoid these limitations, Pan et al. [23] used pure graphite based electrode to detect FRC, where the authors modified the graphite using ammonium carbamate to detect FRC. Maria et al. [24] used a miniaturized carbon black based sensor to detect chlorine dioxide in swimming pool water. Aditya et al. [25] announced a chemical-resistant FRC sensor based on graphene like carbon. However, the electrochemical FRC sensors have not yet been well developed. There is, however, a number of ion-selective transport membranes are available, but very few have been found for HOCl or OCl⁻ because of its strong oxidizing nature.

We also have successfully implemented PG in the aqueous rechargeable Zn/Al ion battery after a simple electrochemical exfoliation process and used DFT calculation to understand the feasibility of $Al^{3+}$ intercalation de-intercalation processes. Graphite/carbon-based rechargeable batteries have been studied extensively that are based on fluorinated natural graphite [26], carbon paper composed of graphite [27], 3D graphitic foam [2, 28], pristine natural graphite [29, 30], highly crystalline graphite flakes [31], and graphene [32]. However, these batteries are based on non-aqueous electrolytes, which are costly. As a cost-effective solution, Angell et al.

[33] proposed molten $AlCl_3$-Urea ionic liquid analog-based energy-storage devices utilizing graphite powder pasted on carbon fiber paper substrate, which is incapable of functioning at room temperature. Inexpensive water-based electrolytes have two times higher ionic conductivity than those of organic electrolytes and can be used to ensure high rate capability and power density [34, 35]. Hence, different methods of using aqueous electrolytes along with the preparation of graphene-based materials more cheaply and efficiently should be investigated further that can allow the use of these electrodes for efficient and environment-friendly energy storage systems [36–38]. Recently our group has used activated carbon as cathode for supercapattery [39] applications, along with nickel phosphate composites as multifunctional electrode platform [4] that include an energy storage device, glucose, and pH sensors.

In the present study, we have investigated developing a FRC sensor based on PG and PDPE in aqueous media. Simple electro-oxidation method has been used for the detection of free residual chlorine. We have extended the work using PG as cathode materials for aqueous Zn/Al ion battery. DFT (density functional theory) calculation has been used to understand the reactions that may happen during the processes. So, these extremely simple, green, and low-cost pencil graphite-derived electrodes can be implemented in the future IOT sensor and energy storage applications.

## Experimental section

### Materials

All the chemicals and the solvents (analytical grade) were used as received in this work. Sodium thiosulfate ($Na_2S_2O_3$), di-sodium hydrogen phosphate ($Na_2HPO_4$), sodium dihydrogen phosphate ($NaH_2PO_4 \cdot 2H_2O$), sodium nitrate ($NaNO_3$), sodium sulfate ($Na_2SO_4$), sodium bicarbonate ($NaHCO_3$), sodium carbonate ($Na_2CO_3$), nitric acid ($HNO_3$) and sodium chloride ($NaCl$) were purchased from Sigma-Aldrich, India. Aluminium chloride ($AlCl_3$), aluminium sulfate ($Al_2(SO_4)_3$), zinc acetate ($Zn(CH_3COO)_2$), hydrochloric acid ($HCl$), potassium chloride ($KCl$), potassium iodide ($KI$), sodium hypochloride ($NaOCl$) were obtained from Merck, Germany. 99.997% dry nitrogen (BOC, Bangladesh) was used for purging purposes. Glass fiber paper separator was purchased from Sigma Aldrich, USA. Cellulose paper was purchased from Viola Vitalis, Dhaka; pencil graphite (Faber Castell pencils 6B) was purchased from Khulna Book Store, Khulna.

### Equipment

Voltammetric, amperometric, and electrochemical impedance spectroscopic (EIS) measurements were carried out using potentiostat/galvanostat with model Biologics SP300. The composition morphology of the samples was examined by SEM (FEI 650 HRSEM Quanta) combined with EDX (INCA energy system, Oxford Instruments) and HRTEM (JEOL HRTEM-2100, 200 kV). The Raman and FT-IR investigations were performed using Renishaw (RA 100) at an excitation of 514.5 nm via confocal Raman Microscope and the Shimadzu IR Tracer-100, respectively.

### Electrochemical measurement as a sensor

PGE was made from commercially available pencil graphite (6B), which was cut into appropriate size and shape (S1a–S1f Fig) and was insulated with an insulating polymer. The fabricated PGE was then used without further modification for the detection of FRC. Afterward, the 6B pencil was used to draw pencil-drawn paper electrode (PDPE) (S1g–S1l Fig) on cellulose paper. Initially, a paper was cut into 5 cm long and drew the electrode within this paper with

the electrode area of 1.25 cm$^2$. After defining the surface area and connection area, PDPE is connected to the potentiostat [40]. Like PGE, the insulating polymer was also applied to PDPE to ensure the defined surface. Biologic SP300 potentiostat was used to measure CV and amperometry. PGE and PDPE were both used as a working electrode, Ag/AgCl as a reference, and platinum wire as a counter electrode in 0.1M PBS. For the interference study, a defined concentration of $NaNO_3$, $NaSO_4$, $NaCO_3$, $NaHCO_3$, and NaCl were used along with NaOCl in 0.1M PBS [41, 42]. Electrode stability was tested using amperometry at an applied potential of 1.5 V vs. Ag/AgCl, with 1 mM of NaOCl added. Electrodes were tested daily over the course of two weeks; and stored at room temperature within a closed glass vial in between the measurements. Prior to subsequent measurements, the electrodes were cleaned in DI water, dried using nitrogen.

## Electrochemical measurement as a battery

The fabricated PGE, as mentioned in the earlier section, was used as a positive electrode after exfoliation. PGE was exfoliated to graphene coated PGE using CV, where Pt wire and Ag/AgCl were used as counter and a reference electrodes, respectively. CV was performed at a scan rate of 50 mV s$^{-1}$ from -1.0 to +1.9 V and repeated at room temperature for 20 cycles in 5.0 M $HNO_3$. Then washed the produced graphene-coated pencil graphite electrode (GPGE) in deionized water to eliminate impurities and dried at room temperature. Finally, GPGE was heated up to 80˚C for one hour in the oven to remove the moisture. In the airtight cell, binder-free GPGE (as prepared) was used as a positive electrode to assemble a battery cell and Zn as a negative electrode in a mixed electrolytes containing zinc acetate (0.5 M) and $AlCl_3$ (0.5 M). Typical mass loading of the cathode material was calculated based on the weight difference measured before and after scrapping the graphene layers from the GPGE. The electrochemical characterization (chronopotentiometry, galvanostatic charging/discharging, and EIS) of bare PGE and GPGE were carried out using a two-electrode cell configuration where GPGE was used as the working electrode and metallic Zn as the counter electrode.

## Computational method

Computational calculations were carried out using Gaussian 16, Revision C.01 series of programs [43]. The energy of geometries were calculated by the DFT-PBEPBE/6-311+G (d,p) level of theory. The unit cell for the graphite sheet adopted a *3 x 3* supercell. The energy was calculated using the Eq (i). On the other hand, the structure of the graphite was fully optimized by DFT-B3LYP/6-311+G (d,p) level of theory. Intercalation of Al and Zn with graphite was also calculated at the same level of theory. The energy was calculated according to Eq (ii).

$$\Delta E(interaction) = E\,(graphite - OCl) - (E\,(graphite) + E\,(OCl) \tag{i}$$

$$\Delta E(interaction) = E\,(graphite - M) - (E\,(graphite) + E\,(M)) \tag{ii}$$

## Results and discussion

### FRC measurement using PGE

To address the analytical applicability of the PGEs, we investigated the electrocatalytic activity of this electrode towards free chlorine in aqueous media. Fig 1(a) shows the CV of PGEs in 0.1M PBS with (red) and without (black) the addition of 20 mM NaOCl. On PGE, a significant increase in anodic current can be observed at around 1.2 V after the addition of NaOCl. It

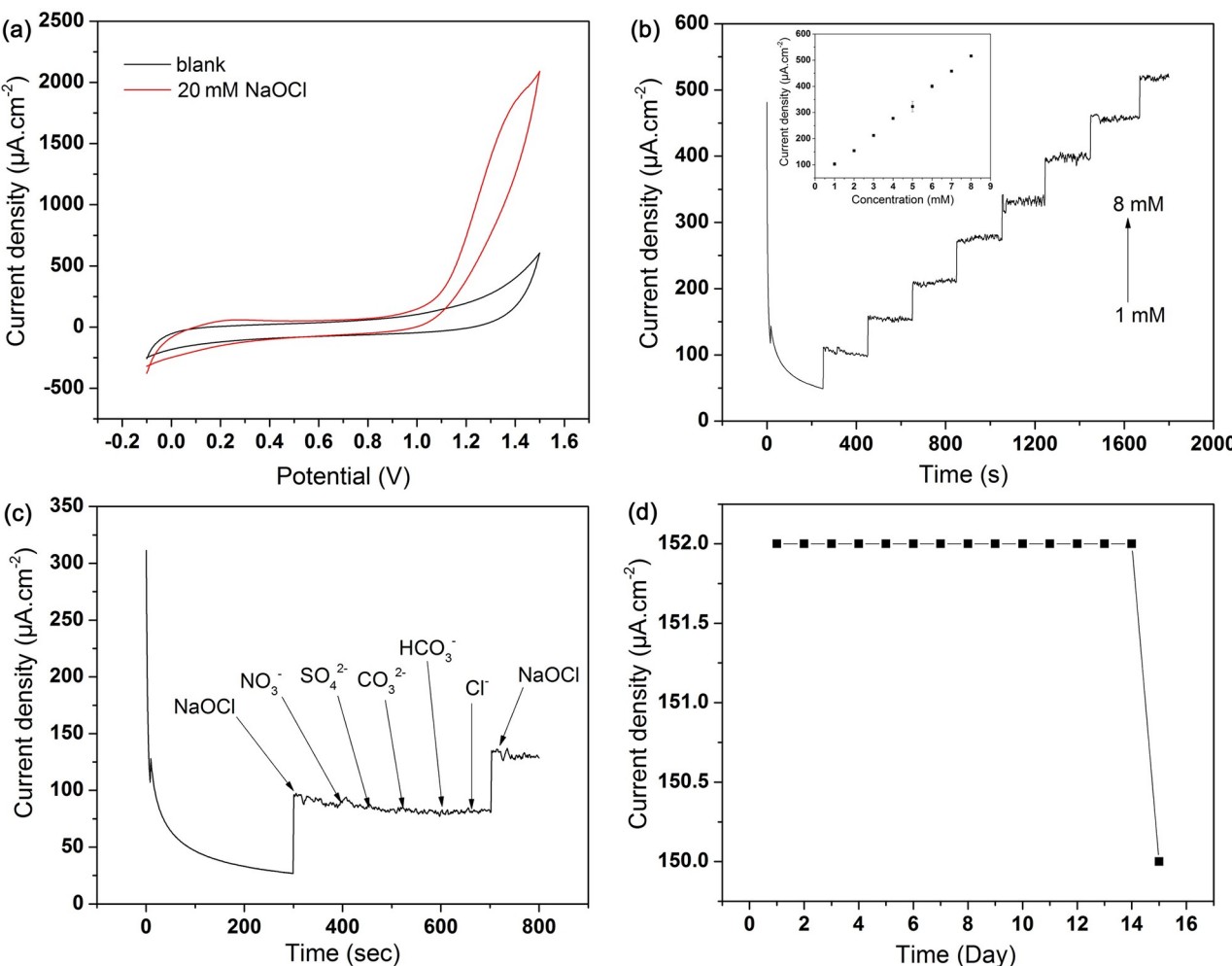

**Fig 1. Electrochemical behaviour of PGE in the presence of NaOCl.** (a) Electro-oxidation on PGE in the presence of 20 mM NaOCl in 0.1 M PBS, scan rate 50 mV s$^{-1}$; (b) amperometric current response on PGE upon successive addition of NaOCl at E$_{app}$ 1.5 V in 0.1 M PBS, corresponding calibration curve (inset); (c) interference study on PGE in the presence of various interferents with 7.3, 13.5, 11.3, 14.5, 1.8 ppm of nitrate, sulfate, carbonate, bicarbonate and chloride ion respectively in 20 mL, 0.1 M PBS (pH 7.4) buffer solution containing 1 mM free chlorine; (d) stability of PGE over two weeks.

specifies that PGEs demonstrate outstanding electro-oxidation current against NaOCl without its surface being modified. These behaviours attribute to the oxidation of graphite in the presence of NaOCl [42]. At oxidizing potential, the graphite instability leads to carbon decomposition into $CO_2$, $HCO_3^-$ and $CO_3^{2-}$. In the presence of hypochlorite (free chlorine), the oxidation of graphite can be significantly accelerated, producing salt and $CO_2$ (Eqs iii & iv) [39]:

$$NaOCl + 3H_2O \rightarrow ClO_3^- + Cl^- + O_2 + H^+ + 6e^- \tag{iii}$$

$$C + 2NaOCl \rightarrow CO_2 + 2NaCl \tag{iv}$$

Reaction mechanism:

$$2H_2O \rightarrow O_2 + 4H^+ + 4e^- \quad E^o = +1.23\ V$$

$$2H_2O + C \rightarrow CO_2 + 4H^+ + 4e^- \quad E^o = +0.144\ V$$

Entwisle [42] also used this electro-oxidation phenomenon of graphite for the chlorine manufacturing by brine electrolysis. Thereby, degradation and stability of the PGE electrode have to be taken into consideration before it can be applied as a free residual chlorine (FRC) sensor. However, the reaction in Eq (iii) mainly happens in the defects and pores that are available on the PGE. Similar behavior was observed for the detection of free chlorine in a boron-doped diamond electrode, where the reaction occurs on the electrode surface [44]. Based on the literature, we postulate Eq (iii), that may occur both on the surface of the PGE as well as in the defects and pores of PGE. Due to this combined effect, the anodic current increased significantly on the addition of NaOCl while PGE as a working electrode. This behaviour cannot be seen when GC, gold, platinum, or any other electrode is used as a working electrode. However, recently Pan et al. [23] used ammonium carbamate modified pencil graphite for FRC detection. Authors mentioned that ammonium carbamate reacts cathodically with NaOCl and increased cathodic current on successive addition of NaOCl in aqueous media.

Fig 1(b) shows the amperometric [45] response by successive additions of 1 mM free chlorine (NaOCl) up to eight additions at the applied potential of 1.5 V. To obtain a better understanding of the sensitivity of the chlorine sensor, the current response changes with concentrations were plotted (Fig 1b inset), where the current changes linearly and resulted in a sensitivity of 50 $\mu A\ mM^{-1}\ cm^{-2}$. The response was repeatable, with a response time less than 3 seconds for a 90% change in signal. Based on the CV and amperometric data and literature, we hypothesized that FRC (NaOCl/OCl$^-$) responds anodically with the PGE at an optimized oxidation potential of +1.50 V. In this experiment, sample de-aeration is not required, as the applied voltage of the amperometric experiments is well outside the voltage range of dissolved oxygen [46]. The limit of detection in these experiments was found to be 46 $\mu M$, which is lower compared to the reported values of 79 and 133 $\mu M$ [47]. In the calibration graph (Fig 1b inset), the standard deviation was found to be negligible (less than 5%) compared to the current obtained for the standard addition. The sensitivity of the PGE sensor for chlorine detection compares well with the reported literature [23, 47, 48]. A brief comparison on the sensitivity of PGE for FRC detection has been shown in S1 Table. Thereby, the PGEs can be used as an alternative material for FRC sensing. Details SEM-EDX and elemental mapping of PGE are shown in S2 and S3 Figs, respectively.

To investigate the selectivity of NaOCl at PGE, the current response for chlorine is calculated at an applied voltage of +1.5 V in stirred 0.1 M PBS solution (pH 7), followed by the introduction of five specific interfering agents [41]. Fig 1(c) indicates the results of potentially interfering contaminants added sequentially (7.3, 13.5, 11.3, 14.5, and 1.8 ppm of sodium nitrate, sodium sulphate, sodium carbonate, sodium bicarbonate, and sodium chloride respectively). The addition of 1 mM NaClO showed a significant current signal as compared to other interfering species. Compared to 1 mM NaClO, the interfering species of nitrate yielded a current response of 4%, sulfate 3%, carbonate 2%, bi-carbonate 1%, and chloride 1%. It confirms the stability of PGE based sensor for detecting FRC in PBS, where up to 97% of the current response can be found up to two weeks (Fig 1d).

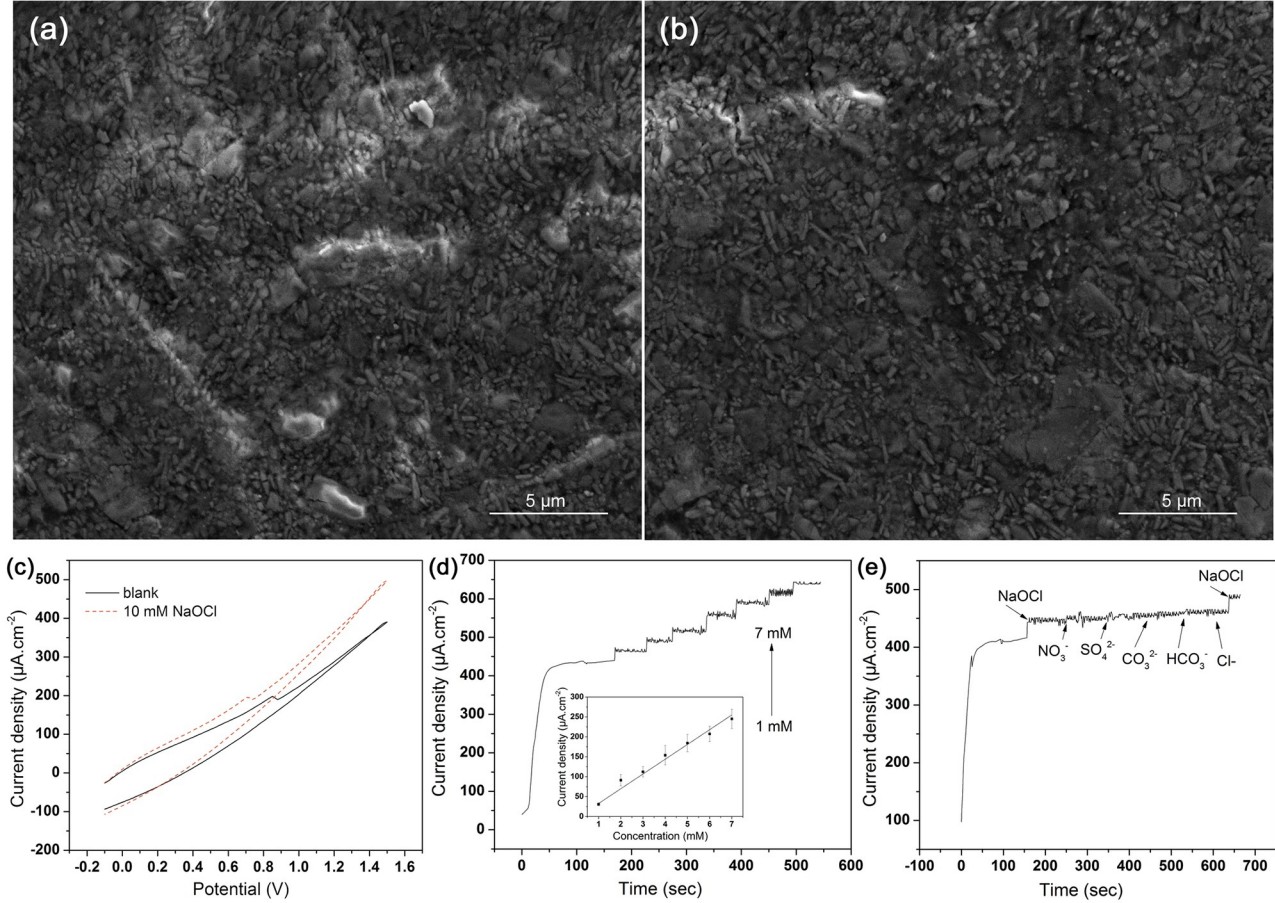

**Fig 2. SEM and Electrochemical behaviour of PDPE in the presence of NaOCl.** (a) SEM of a pencil-drawn paper electrode (PDPE), after drawing one line; (b) SEM of PDPE, after drawing ten lines; (c) Catalytic activity on PDPE in the presence of 10 mM NaOCl in 0.1 M PBS, scan rate 0.05 V s$^{-1}$; (d) amperometric current response on PDPE upon successive addition of NaOCl at $E_{app}$ 1.5 V in 0.1M PBS, corresponding calibration curve (inset); (e) interference study on PDPE in the presence of various interferents with 7.3, 13.5, 11.3, 14.5, 1.8 ppm of nitrate, sulfate, carbonate, bicarbonate and chloride ion respectively at $E_{app}$ 1.5 V in 0.1 M PBS.

### FC measurement using PDPE

Fig 2(a) and 2(b) shows the SEM images of PDPE, with hand-drawn once (Fig 2a) and drawn ten times (Fig 2b). From the images, it can be seen that the amount of graphite on paper increased to ten layers, and the graphite flakes are visible, which are much smaller in size compared to PGE. In this work, we have investigated the electro-analytical activity of PDPE to determine FRC, thereby repeated the whole set of experiment on PDPE. Fig 2(c) presents the CV responses of the PDPE without (black) and with (red) 10 mM NaOCl added in 0.1 M PBS. Same as PGE, a large increase in anodic current can be observed after the addition of NaOCl.

Fig 2(d) demonstrates a typical amperometric response with subsequent addition of 1 mM free chlorine in 0.1 M PBS at an applied potential of 1.5 V on PDPE. Same as PGE, the PDPE electrode could successfully detect FRC at the applied potential of 1.5 V. From the slope (Fig 2d), the sensitivity of free chlorine on PDPE was found to be 27 ± 3.65 µA mM$^{-1}$ cm$^{-2}$ with the limit of detection of 88.9 µM and linearity up to 7 mM. The response time $t_{90}$ for the FRC was less than five seconds. The standard deviation has been calculated for each measurement and shown in Fig 2d inset.

Same as PGE, we have conducted the interference study on PDPE by examining the amperometric responses of 7.3, 13.5, 11.3, 14.5, and 1.8 ppm nitrate, sulfate, carbonate, bicarbonate and chloride ion, respectively in 0.1 M PBS (Fig 2e). The study shows no visible current signal for the addition of sulfate, bicarbonate, carbonate, and nitrate ions, whereas; a high signal can be observed for the addition of FRC. It suggests that PDPE is extremely sensitive and selective towards free chlorine and can be used as a sensor in aqueous media to detect FRC.

## Computational studies

The energy of graphite was calculated by the DFT-PBEPBE/6-311+G (d,p) level of theory. The calculated geometries were shown in Fig 3. The findings revealed that when the oxychloride ion (OCl⁻) interact with the graphite at bond distance 1.43Å, the energy released from the system is -71.93 kcal/mol. On the other hand, when the bond distance is increased gradually and at 2.86Å or above, the energy is absorbed by the system and the magnitude is around 58 kcal/mol. These results are in good agreement with the experimental outcomes shown in Figs 1 and 2. Also, Entwisle proposed the mechanism which has been shown in Eqs (iii) and (iv) satisfied the DFT calculation.

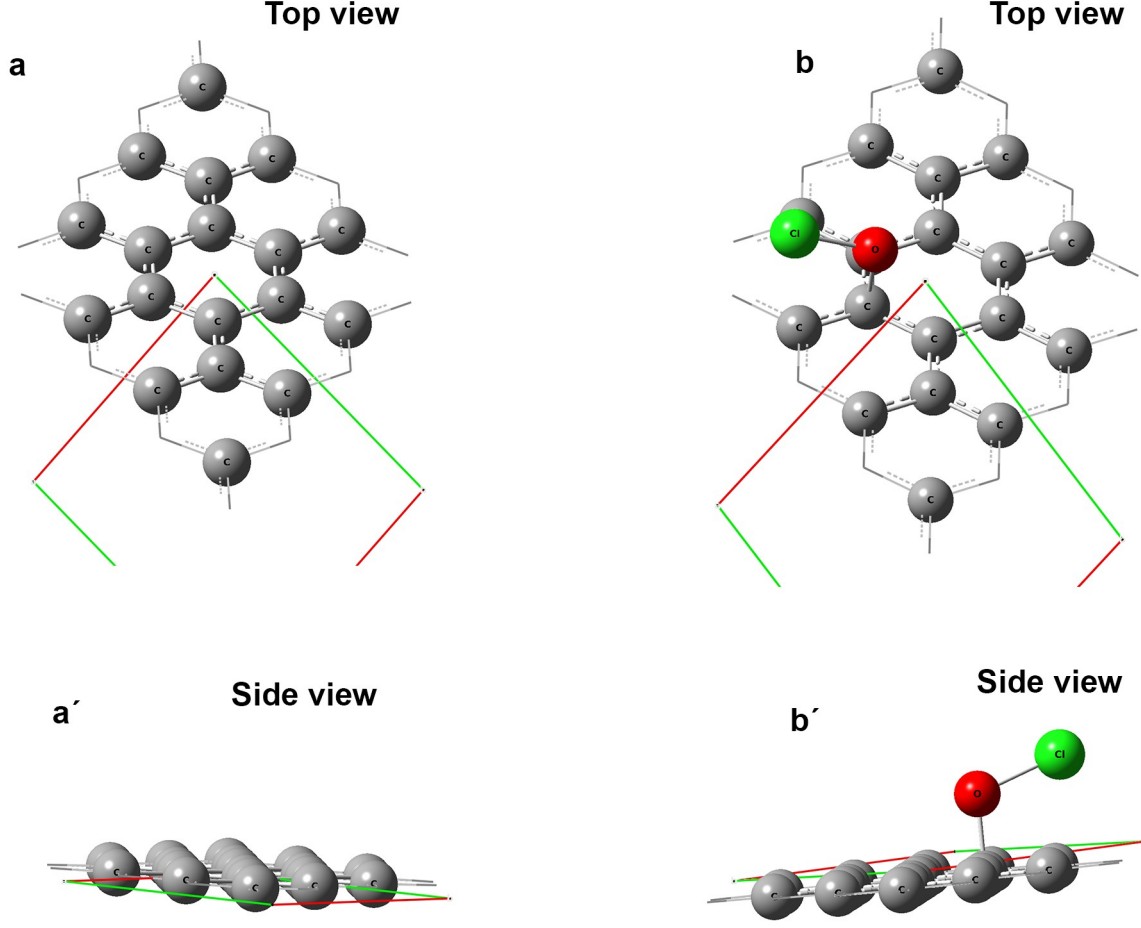

**Fig 3. Geometry of graphene of *3 × 3* supercell.** (a) top view and (a′) side view. Interaction of graphene with oxy-chloride (OCl⁻) anion, (b) top view and (b′) side view.

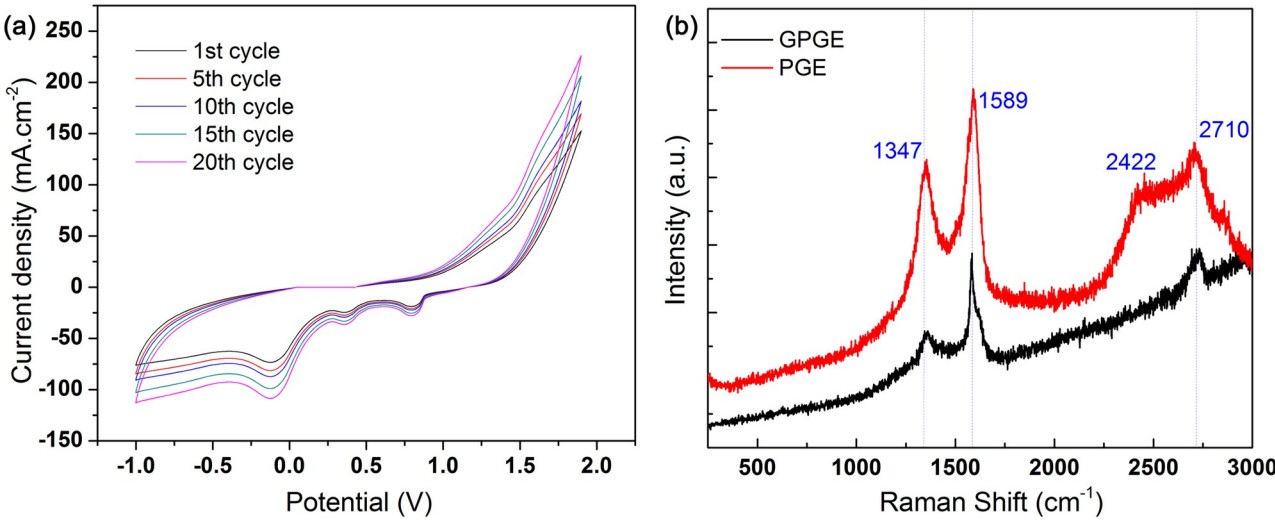

**Fig 4. Fabrication of GPGE and spectral characterization.** (a) Fabrication of GPGE by the multiple cycles (20 cycles) of cyclic voltammogram; condition: PGE as working electrode in 5.0 M HNO₃, scan rate 50 mV/s; (b) Raman Spectrum of PGE and GPGE.

### Fabrication of GPGE for the battery

Graphene preparation on PGE by simple multiple cycles of CV is shown in Fig 4(a). Three clear reduction peaks at +0.70, +0.35, and –0.15 V were observed during the negative potential sweep along with two weak oxidation peaks at +1.25 and +1.64 V during the positive potential sweep in the range of –1.0 to +1.9 V (vs. Ag/AgCl). Fig 4a shows the CV of first, fifth, tenth, fifteenth, and twentieth cycles of graphene synthesis on PGE in 5.0 M HNO₃. The increase in anodic current at +1.25 V represents electrolysis of water by oxidation of $O^{2-}$ ions to $O_2$ gas at PGE (Eq v) [49]. During this stage, the oxygen gas produced causes large forces on the graphitic layer, accompanied by oxidation of the PGE surface, i.e., the formation of oxygen-containing groups (Eq vi). These oxygen-containing groups undergo further oxidation at a higher potential (+1.64 V), causing conversion into graphene oxides on PGE, which can be named in this work as GOPGE (Eq vii) [38].

Reactions during forward potential sweep:

$$2H_2O \rightarrow O_2 + 4H^+ + 4e^- \tag{v}$$

$$4C_x + O_2 \rightarrow 2C_{2x}O \tag{vi}$$

$$(C_{2x}O)_2 + H_2O \rightarrow C_{(2x-2)}COCOOH + H^+ + e^- \tag{vii}$$

Peaks were observed at +0.70, +0.35, and –0.15 V during the reverse potential sweep from +1.9 to –1.0 V. The peak at -0.15 V in this voltammogram was due to the reduction of graphene oxide to graphene on the surface of PGE (Eq viii) [50]. The peaks at +0.70 and +0.35 V can be attributed to the reduction of $NO_3^-$ ions (from HNO₃) to NO and $NO_2$ respectively (Eqs ix and x) [51]. Such gaseous species (NO and $NO_2$) can create extra forces on the graphite

layers, which can increase the interlayer distance between the graphitic or graphene layers.

$$C_{(2x-2)}COCOOH + H^+ + e^- \rightarrow Graphene + H_2O \qquad \text{(viii)}$$

$$NO_3^- + 4H^+ + 3e^- \rightarrow NO + 2H_2O \qquad \text{(ix)}$$

$$NO_3^- + 2H^+ + e^- \rightarrow NO_2 + H_2O \qquad \text{(x)}$$

## Raman and FT-IR of GPGE

Fig 4(b) shows the Raman spectra of PGE and GPGE, where the primary in-plane vibrational mode (G band at ~1589 cm$^{-1}$) is observed along with the second-order overtone of different in-plane vibrational peaks (2D band at ~2710 cm$^{-1}$) of graphene. The 2D band position for both the electrodes shifted slightly. A 9 cm$^{-1}$ Raman shift is found in the GPGE band compared to PGE, partly due to the partial chemical doping [52, 53]. The D bands of both electrodes were determined at 1350 and 1347 cm$^{-1}$ for PGE and GPGE, respectively, where the intensity of D-band ($I_D$) for GPGE is significantly lower than that of PGE. It attributes to the graphene structure being less disordered. From the Raman spectra, it can be seen that both PGE and GPGE has G, 2D, and D bands with different peak intensities and peak maximums, that attributes to the graphene layer on the GPGE surface [38]. The optical properties of the two electrodes were different, and the $I_G$ to $I_D$ ratio for GPGE found to be ~1, which is higher than that for PGE, suggesting the improved optical transparency of the graphene layer on the GPGE [54].

The FT-IR spectra of PGE, and GPGE are displayed in S4 Fig. The small peaks at 2922.16 and 2860.43 cm$^{-1}$ indicate the presence of alkane stretching (from wax) in the graphite structure. The peaks obtained in FT-IR spectra at 3404.36, 3358.07, and 3217.27 cm$^{-1}$ may be due to the vibrations of the oxygen-hydrogen bond (hydroxyl groups) of C–O–H and CO–O–H on the GOPGE surfaces, which is an intermediate stage between PGE and GPGE [55]. At 1736 and 1653 cm$^{-1}$, the peaks of the carbon-oxygen bond vibrations C = O and COOH are visible. Due to the presence of epoxy (C–O–C) groups, the small peak at around 1276.88 cm$^{-1}$ is observed. Peaks obtained from 800–1330 cm$^{-1}$ is also evidence of the presence of C–O groups on the GOPGE [53]. According to FTIR analysis, it can be postulated that in forwarding potential sweep, the PGE surface was oxidized into GOPGE along with the formation of some H$^+$. Then the GOPGE was then again reduced into GPGE in the potential reverse sweep with the production of H$_2$O.

## SEM and TEM of GPGE

SEM and TEM were used to study PGE and GPGE morphologies. The SEM images of PGE with different magnifications (Fig 5a and 5b) and GPGE (Fig 5c and 5d) show changes in the electrode morphologies; sheet-like morphology was observed for the latter case. This SEM analysis supports the synthesis of homogeneous and consistent two dimensional (2D) ultrathin and flexible wrinkled structures of graphene layers on GPGE. Fig 6(a)–6(d) and Fig 6(e)–6(h) show the TEM images of PGE and GPGE, respectively, support the formation of graphene layers. A 2D wrinkled structure of graphene nanosheets of GPGE, with sheet-lengths of 100–125 nm, can be observed in Fig 6(e)–6(h). The HRTEM image in Fig 6(g) show well-resolved lattice fringes of ~0.25 nm, which attributes to high crystallinity of the nanostructures of GPGE as compared to PGE in Fig 6(c). The selected area electron diffraction (SAED) pattern in Fig 6(d) for PGE and 6(h) for GPGE show concentric circles, which are due to the polycrystalline

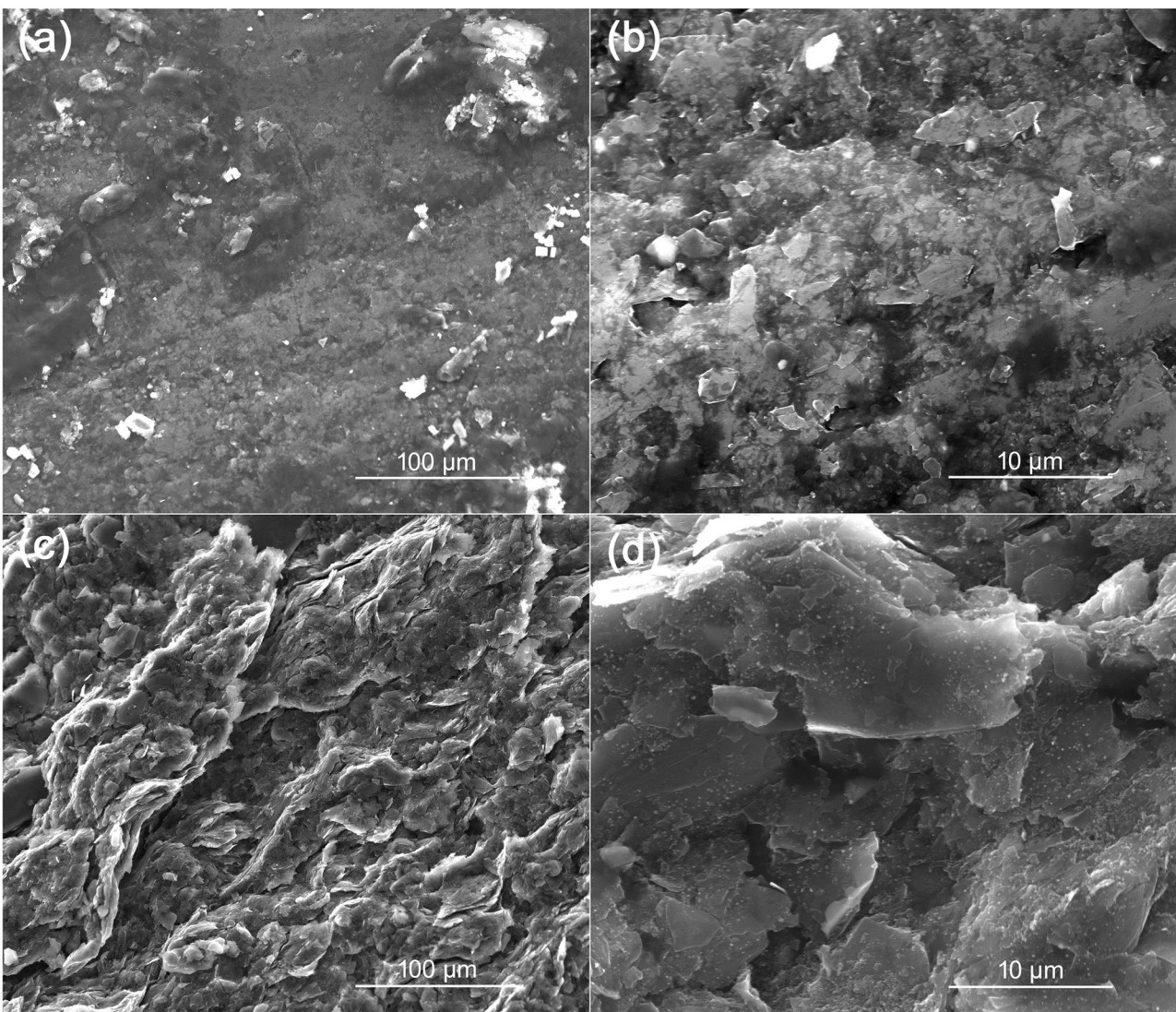

**Fig 5. SEM images of PGE and GPGE.** SEM images of (a-b) PGE and (c-d) GPGE at different magnifications.

nature of both electrodes. However, the GPGE has a relatively regular pattern elucidating a more ordered lattice arrangement in the graphene nanosheets.

### Electrochemical study of GPGE as a battery electrode

In the battery, GPGE was used as a cathode and Zn as an anode with an electrolyte contained both Zn and Al ions. Galvanostatic charge-discharge, cyclic stability, and electrochemical impedance spectroscopy (EIS) were performed in two electrode cell (S5 Fig) configuration. Electrochemical performances of this aqueous Zn/Al-ion battery is shown in Fig 7(a)–7(d). Fig 7a shows an oxidation peak at 1.61 V and reduction peaks at 0.7 and 1.1 V. Due to an increase in over potentials, the difference between cathodic and anodic peak potential increases with the increase in scan rate. The sharp redox peaks are maintained at a high scan rate (10 mV s$^{-1}$) indicating the high rate capability. Conversely, the Zn/Al-ion cell's oxidation peak with PGE as

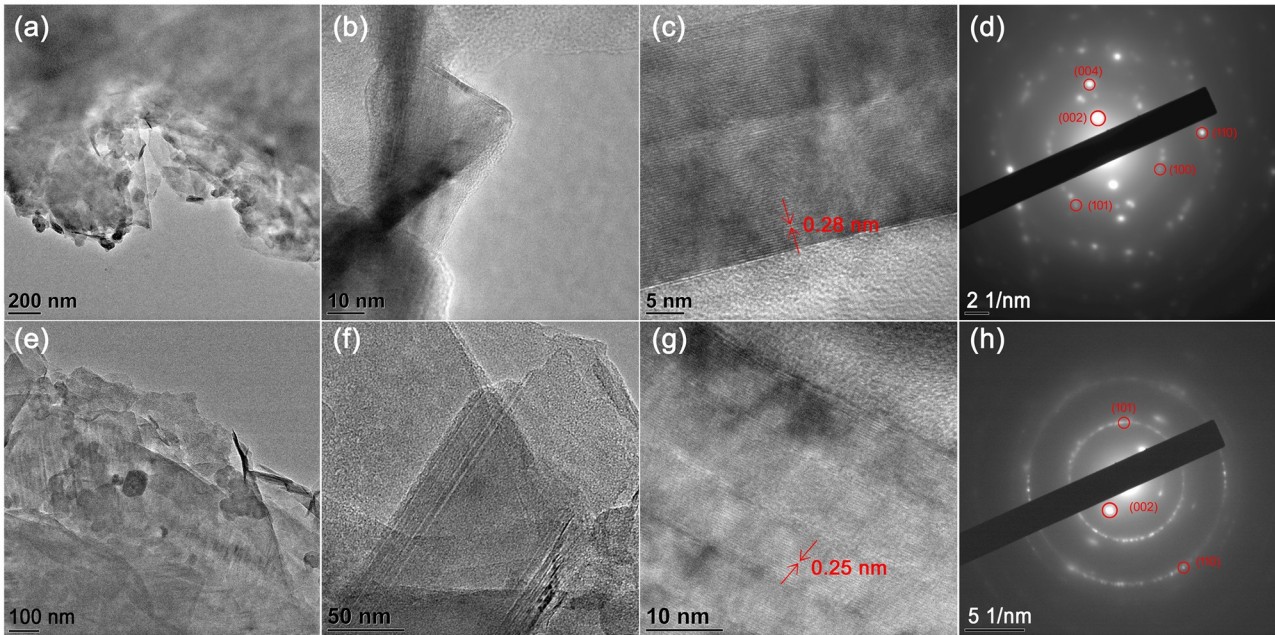

**Fig 6. TEM images of PGE and GPGE.** TEM images of (a-c) PGE with different magnifications and (d) corresponding SAED image; (e-g) TEM images of GPGE with different magnification and (h) corresponding SAED image.

a positive electrode is incomplete with no identifiable oxidation-reduction peaks. The specific capacity of the Zn/Al-ion battery fabricated using GPGE as the positive electrode is determined to be 56.5 mAh g$^{-1}$ at a current density of 1 C (55 mA g$^{-1}$) (Fig 7b). For the GPGE, a maximum discharge capacity of 66.13 mAh g$^{-1}$ with an average discharge voltage of ~1.1 V is achieved at 0.5 C-rate. At 5 C, the Zn/Al ion cell with GPGE as a positive electrode displays a discharge capacity of 38.4 mAh g$^{-1}$. The Zn/Al-ion cell charge-discharge curves for 95 to 100$^{th}$ cycles (Fig 7c) display excellent cyclic stability up to 100 cycles, providing a capacity of 54.1 mAh g$^{-1}$ with capacity retention of ~95.8% at 1 C (Fig 7d). The charging time is also low-enough (~2 hours) for this Zn/Al-ion cell, which is of great aptitude for large–scale stationary energy storage.

The effect of electrolytes has been investigated in the GPGE based battery by running cyclic voltammograms in various solutions of Zn(CH$_3$COO)$_2$, mixed with KCl, HCl and AlCl$_3$ (Fig 8a). A pair of reversible redox peaks can be observed for Zn/Al-ion cells with GPGE in AlCl$_3$/Zn(CH$_3$COO)$_2$ (0.5/0.5 M) solution at +1.51 and +0.78 V, but in case of 0.5 M Zn(CH$_3$COO)$_2$ only, it displays a rectangular CV with little redox properties. In a 1.5 M KCl and 10$^{-4}$ M HCl solutions an asymmetrical CV with no significant oxidation peak is observed, suggesting weak reversibility of redox reactions. It may be inferred from these findings that the redox peaks are increased in the mixed electrolyte of AlCl$_3$ and Zn(CH$_3$COO)$_2$ due to the presence of Al$^{3+}$ or [Al(H$_2$O)$_6$]$^{3+}$ along with Zn$^{2+}$ instead of solo presence of Zn$^{2+}$, H$^+$, CH$_3$COO$^-$, or Cl$^-$ ions. The redox peaks can be recognized as a result of intercalation and deintercalation of Al$^{3+}$ or [Al(H$_2$O)$_6$]$^{3+}$ ions into/from the GPGE, which is similar to the behavior of graphite in ionic liquid electrolyte [34]. During a positive potential sweep, the fabricated electrochemical cell undergoes the charging process using GPGE as a working electrode. Here $Al_xC_{3y}$ gets oxidized by losing 3 electrons and produces $Al^{3+}$ ions deintercalated from graphene $(3C_y)$ (Eqs xi–xiv)

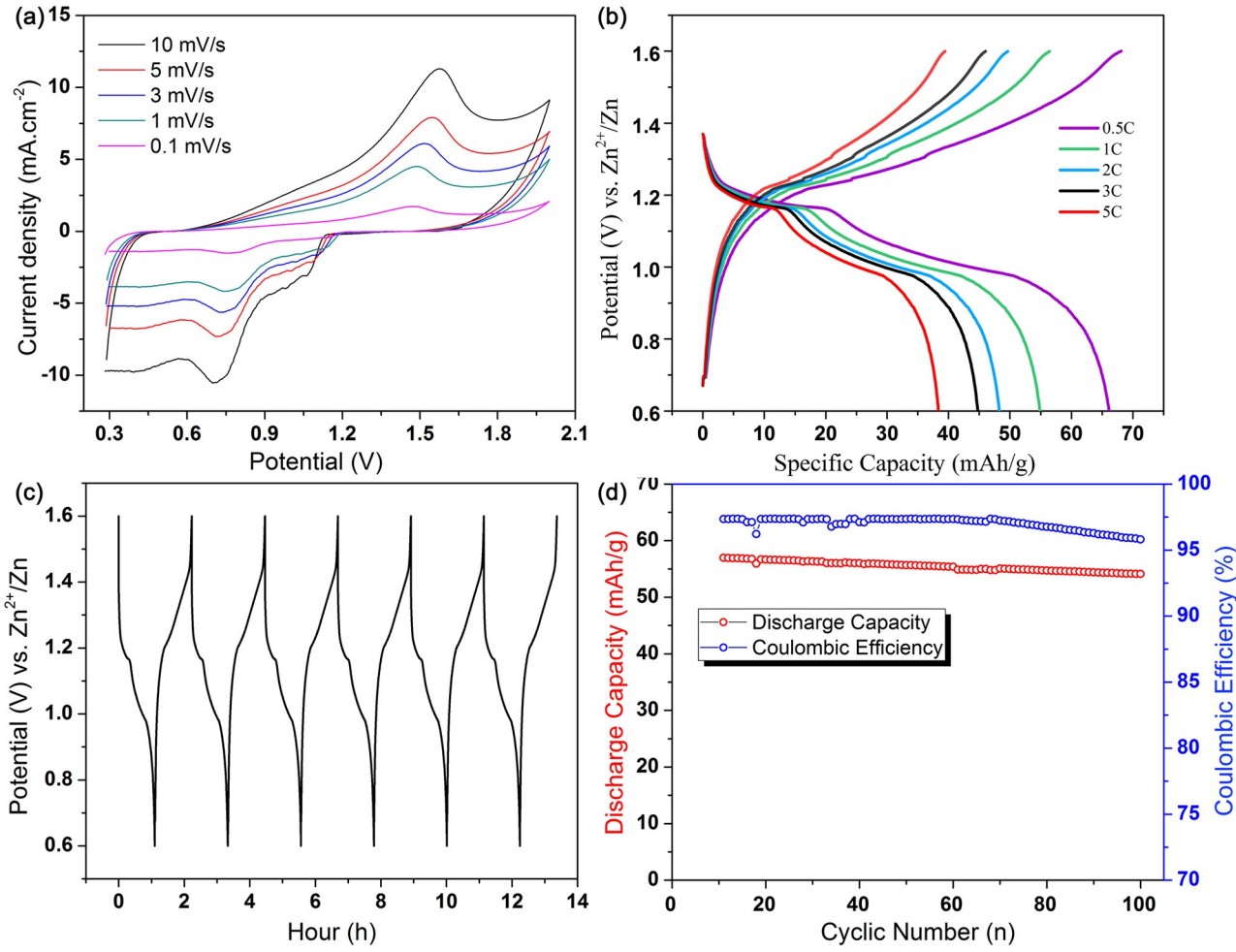

**Fig 7. Charge discharge behaviour of Zn/Al-ion cell.** (a) CV at different scan rates (0.1 to 10 mV/s); (b) Charge/discharge profiles at different C-rates; (c) Chronopotentiometry (95 to 100th cycles); (d) Cycling behavior of Zn/Al-ion cell with GPGE as a positive electrode.

[51].

$$\text{Oxidation (de-intercalation of Al}^{3+}) : Al_xC_y - 3xe^- \rightarrow xAl^{3+} + yC \qquad \text{(xi)}$$

$$\text{Reduction (deposition of Zn)} : Zn^{2+} + 2e^- \rightarrow Zn \qquad \text{(xii)}$$

During a negative potential sweep of CV the cell undergoes discharging process (Eqs x and xi):

$$\text{Reduction (intercalation of Al}^{3+}) : xAl^{3+} + yC + 3xe^- \rightarrow Al_xC_y \qquad \text{(xiii)}$$

$$\text{Oxidation (dissolution of Zn)} : Zn - 2e^- \rightarrow Zn^{2+} \qquad \text{(xiv)}$$

For the Zn/Al-ion cell, the importance of the presence of $Zn^{2+}$ ion and the role of $Al^{3+}$ and $Cl^-$ ions in this electrochemical system is investigated by performing CV in 0.5 M aqueous $AlCl_3$ solution with Zn as an anode and GPGE as cathode. In this case, the CV shows that the

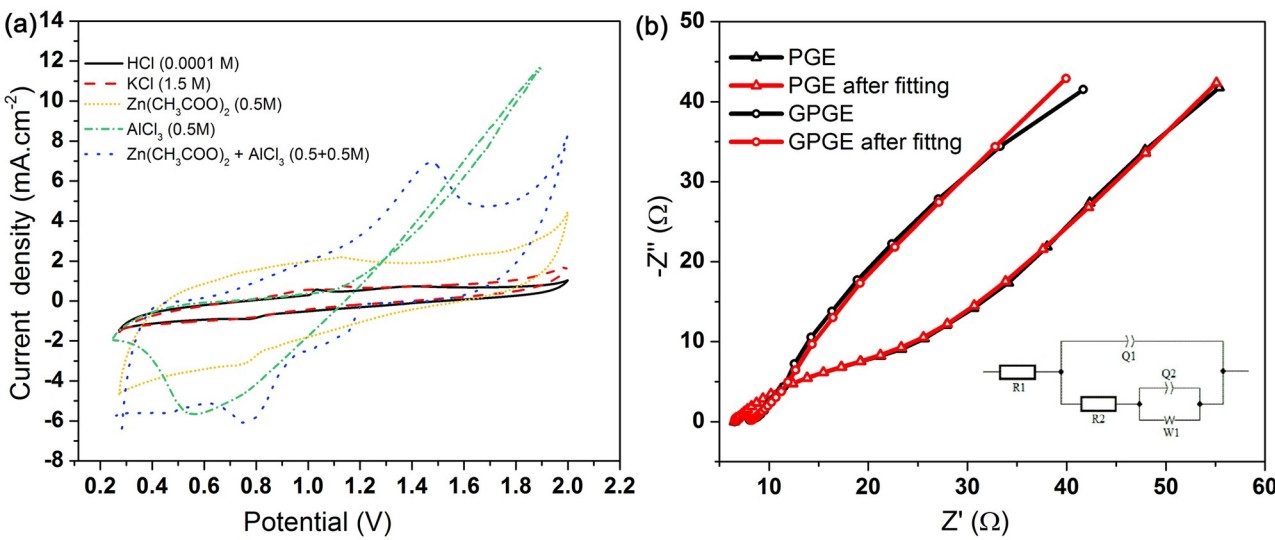

**Fig 8. Effect of electrolytes and Nyquist plot.** (a) CV curves of GPGE in different electrolytes at 1 mV/s; (b) Nyquist plot of EIS for Zn/Al-ion cell with PGE and GPGE as a positive electrode; (inset: equivalent circuit used for fitting EIS).

Zn/GPGE cell can be discharged only where a reduction peak can be observed. The change in the positive current near around +1.2 V is primarily due to the splitting of $H_2O$ into $H^+$ and $OH^-$, which ultimately contributes to the evolution of $O_2$ gas. The absence of oxidation peaks for GPGE verifies that by reducing only available reducible ions ($Al^{3+}$) from the solution, the Zn cannot support the reverse/charging reaction. This phenomenon can be attributed to a lower over-potential for the evolution of hydrogen on aluminum, which makes reducing $Al^{3+}$ ion to pure Al metal in aqueous solution to be incompatible. Therefore, the use of the only $AlCl_3$ as a cell electrolyte makes the fabricated Zn/Al- cell a non-rechargeable one. Thereby, to make the fabricated aqueous cell rechargeable, the presence of metal ions is essential.

Fig 8b shows the EIS curves for GPGE and PGE at open circuit potential (OCP) for Zn/Al-ion battery. The solution resistance ($R_s$) (from the intercept in the real axis) is the same (6.5 Ω) for both PGE and GPGE, because of the same aqueous electrolytes. Nonetheless, the charge–transfer resistance ($R_{ct}$) is determined to be 8.27 Ω for the GPGE, and 33 Ω for PGE, meaning the GPGE will have higher efficiency. The linear region with an angle of about 48˚ at low frequencies shows that the electrode mechanism is both propelled by diffusion and surface controlled. This behavior on the electrode is identical to that of the pseudo-capacitive process. These findings are consistent with that in Fig 6a. At Fig 6a, the anodic current $R^2$ value for the scan rates ranging from 1 to 10 mV s$^{-1}$, is 0.992, suggesting that the Zn/Al-ion cell kinetics using GPGE as the positive electrode interplay between diffusion and surface-controlled reactions, and showed superiority in electrode capacity over pristine PGE.

Ultrathin sheet-like GPGE morphology was observed before Zn/Al-ion cell discharging, which is due to the presence of 2D sp$^2$ carbon layers on the PGE surface (S6a and S6b Fig). On the other hand, granular morphology was obtained after discharging the cell at –0.4V in aqueous $AlCl_3/Zn(CH_3COO)_2$ (0.5/0.5 M) electrolyte (S6c and S6d Fig). These findings suggest Al/Zn intercalation and deposition into/on the cathode (GPGE), followed by electrode surface reduction (Eq xv) [51]:

$$Zn + xAl^{3+} + yC + 3xe^- \ \rightarrow \ Zn^{2+} + Al_xC_y \qquad\qquad (xv)$$

The EDX of GPGE surfaces is shown in (S7 Fig). Analysis of EDX shows that the aluminium content (atomic percentage) increased from 0.82 to 4.41% after deep discharging at −0.4 V. This represents the accumulation or intercalation of aluminum in the GPGE cathode that undergoing reduction. It was also found that the atomic percentage of Zn content increased to 30.16% due to discharging at -0.4 V, where there was practically no presence of Zn before discharge. These results mean that when completely discharged, Zn was deposited only on the outer surface of GPGE. Because of its smaller ionic radius, Al-ion can undergo intercalation within the inner layer of GPGE. In chemical reaction, the reversibility would include two forms of chemistry in charge-discharge cycling in this system: zinc dissolution/deposition ($Zn/Zn^{2+}$), as well as intercalation/deintercalation of Al-ion on GPGE surface. A reasonable rate capability and good cycling stability along with a moderate capacity of fabricated aqueous Zn-Al ion battery attributes to the graphene layer's ultrathin nanostructure that formed on pencil graphite. Electrolytes can access readily to the thin graphene layer, enabling the movement of $Al^{3+}$ ions from bulk to the GPGE surface. Thereby, $Al^{3+}$ ion obtained a larger space (due to enlarged specific surface area) for ion intercalation into the outer graphene layers of GPGEs. Similar findings can be seen in the Li-ion battery [56], Na ion battery [57], and Mg ion battery [58], where $MoS_2$ and $V_2O_5$ with a greater surface area and increased interlayer gap ominously boost ion insertion-extraction kinetics. The average charge and discharge voltages in this system are 1.2 and 1.0 V, respectively, which are similar to aqueous Zn-Al ion batteries using ultrathin graphitic nanosheet [59]. However, it is superior to recently reported systems such as $Al//V_2O_5$ (average voltages of 0.8 and 0.6 V for charge and discharge) [60] and $Al//VO_2$ (average voltages of 0.7 and 0.5 V for charge and discharge) [61] in an ionic liquid. The specific capacity for this work (56.46 mAh $g^{-1}$) is smaller than that of using an ionic liquid electrolyte. It is nevertheless more proficient in terms of manufacturing costs and eases compared with pyrolytic graphite-$AlCl_3$/EMIC electrolyte system (<70 mAh $g^{-1}$) [2], the chloride ion-ionic liquid electrolyte system [62], $TiO_2$ nanotubes based device [63], Prussian blue analog nanoparticles with $AlCl_3$ and $Al(NO_3)_3$ in aqueous electrolytes based system [64, 65]. From the computational study (S8 Fig), it can be seen that intercalation energy of Al is -415.43 kcal/mol, whereas the energy of Zn is -104.42 kcal/mol. It reveals that Al intercalation with graphite is about 4 times favourable compared to intercalation of Zn. These results are in good agreement with the experimental outcomes shown in Eqs (viii) to (xi).

Compared to most Al ion batteries in an ionic liquid and aqueous electrolytes, the cycling behaviors considered excellent (95.8% up to 100 cycles) (S2 Table). Zn is also a low cost, non-toxic material produced on a large scale of approximately 12 million tons annually [66]. The formation of the dendrite is usually a limitation against rechargeable Zn electrodes, which have not been observed within 100 cycles in this system. Another issue before the practical implementation of this aqueous rechargeable Zn/Al-ion battery is that it requires to use a highly concentrated Al salt electrolyte to achieve sufficient energy density, comparable to dual–graphite cells [67]. However, it needs to keep in mind that due to the highly acidic in nature concentrate aluminium salt-based electrolyte can corrode the Zn electrode.

## Conclusions

This study has identified PGE as a low-cost, easy-to-manufacture sensing platform for detecting FRC in aqueous media. This sensor exhibited a stable response and good selectivity in the presence of interfering ionic species. The PGE shows a sensitivity of 50 μA $mM^{-1}$ $cm^{-2}$ with a wide linearity range. The stability of PGE as the FRC sensor is ~97% after 14 days of its first activity. The response is linear, with response time ($t_{90}$) less than 3 s for the FRC detection. We have successfully applied this system to paper and made a flexible PDPE for FRC detection. At

the same time, we have modified and exfoliated this PGE by a simple CV and formed a graphene layer on the PGE core rod and successfully implemented as a cathode for an aqueous Zn/Al ion battery. The high specific surface area of these ultrathin graphene layers provides sufficient space for the intercalation of Al-ion as well as for the deposition of Zn on the GPGE's outer surface. The rechargeable battery can thus give a practical ability while maintaining a reasonable current density during processes of charging and discharging. After 100 cycles, Zn dendrite is not observed on the anode. However, during cycling, small volume expansion of GPGE is observed, which can cause disintegration of the cathode and limit the cycle life. Also, $AlCl_3$-electrolyte's high acidity is prone to corrode the Zn metal, which may result in self-discharging and loss of capacity. Proactive nature of anodic materials (e.g., hyper dendritic nanoporous 3D-Zn foam as an anode), 3D-Nickel foam surface stabilized graphene layers as cathodic material, together with electrolytes produced from Al salt solution may be used to solve such inherent problems.

## Supporting information

**S1 Fig. PGE and PDPE fabrication.** Stepwise fabrication of PGE (a-f), and PDPE (g-l).
(TIF)

**S2 Fig. SEM-EDX of PGE.** (a) SEM of PGE; (b) Composition on the PGE surface; (c) EDX of PGE.
(TIF)

**S3 Fig. Elemental mapping of PGE.** (a-f) EDX elemental mapping of PGE.
(TIF)

**S4 Fig. FT-IR spectra of PGE, GOPGE and GPGE.** FT-IR of PGE, GOPGE and GPGE, CV was used to exfoliate PGE to GPGE, where Pt wire and Ag/AgCl were used as counter and reference electrode respectively. CV was performed at a scan rate of 50 mV s$^{-1}$ from -1.0 to +1.9 V and repeated at room temperature for 20 cycles in 5.0 M $HNO_3$.
(TIF)

**S5 Fig. Schematic diagram of Zn/Al ion battery.** Schematic diagram of Zn/Al ion battery, where Zn is used as anode and GPGE is used as cathode and $AlCl_3/Zn(CH_3COO)_2$ (0.5/0.5 M) is used as electrolytes.
(TIF)

**S6 Fig. SEM images of GPGE.** SEM images of GPGE before (a-b) and after (c-d) discharging to -0.4V (vs. Ag/AgCl).
(TIF)

**S7 Fig. Elemental mapping of GPGE before and after discharge.** SEM and elemental mapping of GPGE before discharging (a-c) and after discharging (d-f) the cell at -0.4V (vs. Ag/AgCl).
(TIF)

**S8 Fig. Intercalation of graphite 16×16 carbon atom.** Intercalation of graphite with Al (a) top view and (a′) side view; interaction of graphite 16×16 carbon atom with Zn (b) top view and (b′) side view.
(TIF)

**S1 Table. Various FRC sensors.** Comparison on the performance of different electrochemical FRC sensors.
(DOCX)

**S2 Table. Graphite electrode based batteries.** Comparison on efficiencies of different battery based on graphite electrodes.
(DOCX)

**S1 TOC.**
(TIF)

## Acknowledgments

The authors thank Mr. Apu K. Dutta (KUET) and Mr. Juel Islam (KUET) for their contribution.

## Author Contributions

**Conceptualization:** Md. Mizanur Rahman Badal, Kafil M. Razeeb, Mamun Jamal.

**Data curation:** Jahidul Islam, Han Shao, Md. Mizanur Rahman Badal, Mamun Jamal.

**Formal analysis:** Jahidul Islam, Han Shao, Kafil M. Razeeb, Mamun Jamal.

**Funding acquisition:** Mamun Jamal.

**Investigation:** Jahidul Islam, Mamun Jamal.

**Methodology:** Jahidul Islam, Mamun Jamal.

**Project administration:** Mamun Jamal.

**Resources:** Mamun Jamal.

**Software:** Han Shao, Md. Mizanur Rahman Badal, Kafil M. Razeeb, Mamun Jamal.

**Supervision:** Mamun Jamal.

**Validation:** Han Shao, Md. Mizanur Rahman Badal, Kafil M. Razeeb, Mamun Jamal.

**Visualization:** Mamun Jamal.

**Writing – original draft:** Jahidul Islam, Mamun Jamal.

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
