## [Decision Letter · Decision Letter 0]

15 Dec 2020

PONE-D-20-33482

Pencil graphite as electrode platform for free chlorine sensors and energy storage devices

PLOS ONE

Dear Dr. Jamal,

Thank you for submitting your manuscript to PLOS ONE. After careful consideration, we feel that it has merit but does not fully meet PLOS ONE’s publication criteria as it currently stands. Therefore, we invite you to submit a revised version of the manuscript that addresses the points raised during the review process.

We look forward to receiving your revised manuscript.

Kind regards,

Zafar Khan Ghouri

Academic Editor

PLOS ONE

Journal Requirements:

2. Please ensure that vendor details for all materials and equipment used are provided. We note that these may be missing for the cellulose paper and pencil graphite.

Reviewers' comments:

Reviewer's Responses to Questions

**Comments to the Author**

1. Is the manuscript technically sound, and do the data support the conclusions?

Reviewer #1: Yes

Reviewer #2: Partly

2. Has the statistical analysis been performed appropriately and rigorously? 

Reviewer #1: Yes

Reviewer #2: Yes

3. Have the authors made all data underlying the findings in their manuscript fully available?

Reviewer #1: Yes

Reviewer #2: Yes

4. Is the manuscript presented in an intelligible fashion and written in standard English?

Reviewer #1: Yes

Reviewer #2: No

5. Review Comments to the Author

Reviewer #1: This paper reports the use of pencil graphite as an electrode for

dual applications that include the detection of free residual chlorine using electro-oxidation process

and as an electrochemical energy storage cathode. The pencil graphite is then transferred to

cellulose paper by drawing ten times and applied for the detection of free residual chlorine, which

shows a sensitivity of 27 µA mM-1

cm-2 with a limit of detection of 88.9 µM and linearity up to 7

mM

the work is performed systematically and it is presented nicely in the manuscript.

Pencil graphite can be used for both sensor and energy-storage

applications

Reviewer #2: The authors have studied and reported the use of Pencil Graphite as Electrode Platform for Free Chlorine Sensors and Energy Storage Devices. While I greatly appreciate the authors for the effort taken, I feel the manuscript needs major revision to be accepted for publication.

1. Incorrect sentence formation is present at various parts of the manuscript that makes the readability of manuscript difficult. The authors need to seriously look into it.

2. The cyclic stability test was carried out only for 100 cycles and it shows a retention of 95.8%. This shows the poor efficiency of the material. The authors are recommended to carry out the test for higher number of cycles (>5000 at least).

3. In the introduction, various literatures are written (ex: Mariad, Maria et al. Aditya et al.) However, the citations to the same are not present.

4. In the materials section, it is advised to mention the name of the compounds and then write their chemical formula, instead of writing only the formula.

5. The authors claim that synthesized pencil graphite is a green material. The authors have to justify this statement.

6. The full form of PDPE to be abbreviated initially. Correct all such mistakes.

7. ‘Every day the electrode was tested for two weeks.’ No meaning and connection to the previous statement.

8. Caption of Fig. 2 appears in the middle of a paragraph, which breaks the continuity.

9. The authors could quantify FRC using UV spectroscopy and other analytical techniques for better comparison.

6. PLOS authors have the option to publish the peer review history of their article (what does this mean?). If published, this will include your full peer review and any attached files.

Reviewer #1: **Yes: **Pratap Kollu

Reviewer #2: No

---

## [Author Response · Author response to Decision Letter 0]

23 Jan 2021

PONE-D-20-33482

Pencil graphite as electrode platform for free chlorine sensors and energy storage devices

The authors would like to thank the reviewers for their valuable comments and suggestions to improve the manuscript. The response to the reviewers are as follows:

Reviewer #1: This paper reports the use of pencil graphite as an electrode for dual applications that include the detection of free residual chlorine using electro-oxidation process and as an electrochemical energy storage cathode. The pencil graphite is then transferred to cellulose paper by drawing ten times and applied for the detection of free residual chlorine, which shows a sensitivity of 27 µA mM-1 cm-2 with a limit of detection of 88.9 µM and linearity up to 7

mM the work is performed systematically and it is presented nicely in the manuscript. Pencil graphite can be used for both sensor and energy-storage applications.

Responses: Thanks to the reviewer for appreciating our work.

Reviewer #2: The authors have studied and reported the use of Pencil Graphite as Electrode Platform for Free Chlorine Sensors and Energy Storage Devices. While I greatly appreciate the authors for the effort taken, I feel the manuscript needs major revision to be accepted for publication.

Responses: We would like to thank the reviewer for the critical analysis of our manuscript. Revisions have been made in the updated version of the manuscript.

1. Incorrect sentence formation is present at various parts of the manuscript that makes the readability of manuscript difficult. The authors need to seriously look into it.

Responses: English has been corrected throughout the manuscript. A copy of the manuscript with track changes has been enclosed with the submission.

2. The cyclic stability test was carried out only for 100 cycles and it shows a retention of 95.8%. This shows the poor efficiency of the material. The authors are recommended to carry out the test for higher number of cycles (>5000 at least).

Responses: In this work, we have showed Zn as anode, graphene coated PGE as cathode and a mixture of aluminum chloride (AlCl3) and zinc acetate (Zn(CH3COO)2) as electrolyte. In terms of safety, simplicity and cost, the proposed aqueous battery possess a huge potential as a cost effective energy storage device. However, we have only studied the cyclic stability test up to 100 cycles, which shows almost no efficiency loss. Currently, work is in progress to improve the cyclability of this battery. As these experiments are on-going, the outcome of these studies will be reported in a separate article. It is also worth mentioning that Angell et al. [1] fabricated a graphite powder based cathode material based aqueous battery with Al3+ as electrolyte, and reported the cyclability up to 180 cycles. Similar graphitic materials based aqueous energy storage devices have been reported by many other authors [2-6] with similar cycle life. 

References

1. Angell, M.; Pan, C.-J.; Rong, Y.; Yuan, C.; Lin, M.-C.; Hwang, B.-J.; Dai, H., High Coulombic efficiency aluminum-ion battery using an AlCl3-urea ionic liquid analog electrolyte. Proceedings of the National Academy of Sciences 2017, 114 (5), 834-839.

2. Kravchyk, K. V.; Wang, S.; Piveteau, L.; Kovalenko, M. V., Efficient Aluminum Chloride–Natural Graphite Battery. Chemistry of Materials 2017, 29 (10), 4484-4492.

3. Sun, H.; Wang, W.; Yu, Z.; Yuan, Y.; Wang, S.; Jiao, S., A new aluminium-ion battery with high voltage, high safety and low cost. Chemical Communications 2015, 51 (59), 11892-11895.

4. Wang, W.; Jiang, B.; Xiong, W.; Sun, H.; Lin, Z.; Hu, L.; Tu, J.; Hou, J.; Zhu, H.; Jiao, S., A new cathode material for super-valent battery based on aluminium ion intercalation and deintercalation. Scientific Reports 2013, 3 (1), 3383.

5. Lin, M.-C.; Gong, M.; Lu, B.; Wu, Y.; Wang, D.-Y.; Guan, M.; Angell, M.; Chen, C.; Yang, J.; Hwang, B.-J.; Dai, H., An ultrafast rechargeable aluminium-ion battery. Nature 2015, 520 (7547), 324-328.

6. Zhang, X.; Tang, Y.; Zhang, F.; Lee, C.-S., A Novel Aluminum–Graphite Dual-Ion

Battery. Advanced Energy Materials 2016, 6 (11), 1502588.

3. In the introduction, various literatures are written (ex: Mariad, Maria et al. Aditya et al.) However, the citations to the same are not present.

Responses: It is fixed in the manuscript.

4. In the materials section, it is advised to mention the name of the compounds and then write their chemical formula, instead of writing only the formula.

Responses: It is fixed in the manuscript.

5. The authors claim that synthesized pencil graphite is a green material. The authors have to justify this statement.

Responses: In both the abstract and introduction section, we have claimed about the green materials as follows:

Abstract

“Therefore, pencil graphite due to its excellent electro-oxidation and conducting properties, can be successfully implemented as a low cost, disposable and green material for both sensor and energy-storage applications.”

Introduction

“So, these extremely simple, green, and low-cost pencil graphite-derived electrodes can be implemented in the future IOT sensor and energy storage applications.”

Please find below the justification of PGE as green material:

Green material involves eliminating and avoiding the use of toxic and hazardous reagents and solvents in the manufacturing and applications process [1, 2]. Pencil graphite is a biodegradable material and also recyclable. These commercial pencil graphite does not use any hazardous chemicals. In addition, using pencil drawn on to paper make the fabrication more environment friendly. In our case, we used the pencil graphite without any further modification as free residual chlorine sensor, and with a simple one-step cyclic voltammetry, it has been used as cathode in aqueous battery application. Thereby, it was termed as a green material.

Reference

1. Srinivasulu Kanaparthi and Sushmee Badhulika, Solvent-free fabrication of a biodegradable all-carbon paper based field effect transistor for human motion detection through strain sensing, Green Chem., 2016,18, 3640-3646.

2. https://en.wikipedia.org/wiki/Green_chemistry.

6. The full form of PDPE to be abbreviated initially. Correct all such mistakes.

Responses: These type of issues have been fixed in the manuscript.

7. ‘Every day the electrode was tested for two weeks.’ No meaning and connection to the previous statement.

Responses: Following lines are included now in the manuscript to make the connections with the statements.

“The electrode stability was tested using amperometry at an applied potential of 1.5 V vs. Ag/AgCl, with 1 mM of NaOCl added. Electrodes were tested daily over the course of two weeks; and stored at room temperature within a closed glass vial in between the measurements. Between subsequent measurements, the electrodes were cleaned in DI water and dried using nitrogen gas.”

8. Caption of Fig. 2 appears in the middle of a paragraph, which breaks the continuity.

Responses: It is fixed in the manuscript.

9. The authors could quantify FRC using UV spectroscopy and other analytical techniques for better comparison.

Responses: Currently we are working on inter laboratory and intra laboratory validation of FRC sensor, along with substrate improvement. We are hoping to publish these work in a separate paper, as currently it is in progress. As this work is the first of its kind, we have studied the behavior of PGE and PDPE as FRC sensor, as well as showed its feasibility as energy storage materials with small modification. Currently, work is in progress to improve the nature of the substrate so that these electrodes can be used in any real samples.

---

## [Editor Report · Decision Letter 1]

22 Feb 2021

Pencil graphite as electrode platform for free chlorine sensors and energy storage devices

PONE-D-20-33482R1

Dear Dr. Jamal,

We’re pleased to inform you that your manuscript has been judged scientifically suitable for publication and will be formally accepted for publication once it meets all outstanding technical requirements.

Kind regards,

Zafar Khan Ghouri

Academic Editor

PLOS ONE

Additional Editor Comments (optional):

It's my pleasure to inform you that, after the peer review, your paper, Pencil graphite as electrode platform for free chlorine sensors and energy storage devices has been accepted.
---

## [Editor Report · Acceptance letter]

25 Feb 2021

PONE-D-20-33482R1 

Pencil graphite as electrode platform for free chlorine sensors and energy storage devices 

Dear Dr. Jamal:

I'm pleased to inform you that your manuscript has been deemed suitable for publication in PLOS ONE. Congratulations! Your manuscript is now with our production department. 

Kind regards, 

on behalf of

Dr. Zafar Ghouri 

Academic Editor

PLOS ONE